# First Isolation and Identification of Homologous Recombination Events of Porcine Adenovirus from Wild Boar

**DOI:** 10.3390/v14112400

**Published:** 2022-10-29

**Authors:** Mami Oba, Sumiya Borjigin, Fuka Kikuchi, Toru Oi, Hitoshi Takemae, Hiroho Ishida, Hironobu Murakami, Naoyuki Aihara, Takanori Shiga, Junichi Kamiie, Tetsuya Mizutani, Makoto Nagai

**Affiliations:** 1Research and Education Center for Prevention of Global Infectious Diseases of Animals, Tokyo University of Agriculture and Technology, Fuchu 183-8509, Tokyo, Japan; 2School of Veterinary Medicine, Azabu University, Sagamihara 252-5201, Kanagawa, Japan; 3Center for Surveillance, Immunization, and Epidemiologic Research, National Institute of Infectious Diseases, Tokyo 162-8640, Japan; 4Faculty of Bioresources and Environmental Science, Ishikawa Prefectural University, Nonoichi 921-8836, Ishikawa, Japan

**Keywords:** porcine adenovirus, wild boar, isolation, homologous recombination, protein structure

## Abstract

Porcine adenoviruses (PAdVs) are distributed in pig populations and classified into five immunologically distinct serotypes (PAdV-1 to 5). In this study, a PAdV was isolated from a fecal sample of wild boar for the first time. Whole-genome analysis revealed that this strain (Ino5) has sequence homology (approximately > 93%) throughout the genome with the PAdV-5 strain HNF-70 that was isolated from a pig in Japan in 1987, except for the hexon, E3 612R, and fiber coding regions. Two possible recombination breakpoints were detected in the hexon and E3 612R regions, which were found to have reduced GC content. Structural prediction analysis showed that a part of the hexon protein corresponding to the tower region of Ino5 had structural differences when compared with HNF-70, suggesting antigenic heterogeneity between these strains. PAdVs were detected in 1.77% (2/113) and 12% (12/100) of the fecal samples from wild boars and pigs collected in Japan by PCR, respectively. Phylogenetic analyses of the hexon and fiber genes revealed that some samples showed different grouping in the hexon and fiber genes, suggesting that these viruses have recombination events. These findings suggest that the PAdV-5 has evolved with homologous recombination events in the same manner as human adenoviruses among not only pig populations, but also wild boars in Japan.

## 1. Introduction

Adenoviruses are linear double-stranded DNA viruses that have an icosahedral non-enveloped structure including three major proteins, i.e., the penton base, hexon, and fiber. At present, they are divided into six genera: *Atadenovirus*, *Aviadenovirus*, *Ichtadenovirus*, *Mastadenovirus*, *Siadenovirus*, and *Testadenovirus*, based on the phylogenetic distance of the amino acid (aa) sequence of the polymerase gene within the family *Adenoviridae* [1]. Porcine adenoviruses (PAdVs) belong to the genus *Mastadenovirus*. Cattle, sheep, and humans can be infected not only by mastadenoviruses, but also by viruses of the atadenovirus, whereas atadenovirus infection of swine has not been reported. PAdVs are classified into five immunologically distinct serotypes named PAdV 1 to 5 [2]. Species demarcation criteria in the genus *Mastadenovirus* is currently being replaced by criteria for species demarcation based on evolutionary distance [1]. According to this demarcation, PAdV 1 to 3, 4, and 5 were reclassified into porcine mastadenovirus A, B, and C, respectively. PAdV (PAdV-1) was first isolated from a rectal swab obtained from a 12-day-old pig with diarrhea in the 1960s, followed by PAdV-2 and -3 from rectal swabs collected from healthy pigs [3,4]. PAdV-4 was isolated from the brain of a 10-month-old pig showing incoordination, anorexia, and enteritis [5]. In 1987 in Japan, PAdV-5 strains HNF-61 and HNF-70 were isolated from nasal swabs of fattening pigs with respiratory symptoms [6]. Additional genotype PAdVs isolated from spleen lymphocytes and urothelial cells from pigs without clinical symptoms have been reported [7,8]. Although PAdVs are thought to be prevalent in the pig-raising industry throughout the world [9,10,11,12], identification of PAdV from wild boar has not been reported. Furthermore, at present, only one complete genome sequence of PAdV-5 was available in the GenBank database [13]. In this study, we isolated a PAdV from wild boar hunted in 2020 for the first time. Complete genome analysis reveals that the PAdV has sequence homology to the PAdV-5 strains, and homologous recombination events among PAdVs were identified. Furthermore, the prevalence of PAdV were investigated using fecal samples from wild boars and pigs collected between 2016 to 2021 in Japan.

## 2. Materials and Methods

### 2.1. Virus Isolation from Fecal Samples of Wild Boar

During enteric virus surveillance using fecal samples of wild boar, virus isolation was performed using a cloned porcine kidney (CPK) [14,15], Vero KY5 [16], BHK-21 (JCRB9020), and MDBK (JCRB9028) cell lines. Rectum contents obtained from hunted wild boars in Japan in 2020 were diluted 1:9 (*w/v*) in Eagle minimum essential medium (MEM) containing 0.075 mg of sodium bicarbonate per ml, 0.292 mg of L-glutamine per ml, and centrifuged at 1500× *g* for 10 min, followed by filtration through 0.45 and 0.22 µm filters. The filtrated samples were activated by adding an equal volume of 20 µg/mL trypsin dissolved in the MEM prior to inoculation. The activated sample was diluted twofold with MEM, and 100 µL of that was inoculated onto confluent monolayers of cells in 24 well plates. The cells were washed with PBS three times before inoculation. After a 1 h incubation at 37 °C for adsorption, 200 µL of Opti-MEM was added to each well. For the CPK cells, 200 µL of MEM containing 5% FBS was added the day after inoculation. If cytopathic effect (CPE) did not appear after 7 days of incubation, the subsequent passages were carried out using supernatants in the same manner.

### 2.2. Construction of DNA and cDNA Libraries and Deep Sequencing

Total nucleic acid of the cultures of the supernatant of the cells was extracted using High Pure Viral Nucleic Acid Kit (Roche Molecular Systems Inc., Pleasanton, CA, USA.) according to manufacturer’s instructions. Subsequently, a DNA and cDNA library for next generation sequencing (NGS) was carried out using the Nextera XT DNA Library Preparation Kit (Illumina, San Diego, CA, USA) and the NEBNext Ultra II RNA Library Prep Kit for Illumina (New England Biolabs, Ipswich, MA, USA), respectively. NGS was performed on a MiSeq benchtop sequencer (Illumina, San Diego, CA, USA) using 150 paired-end reads. Sequence data in FASTQ format were retrieved from the MiSeq Reporter (Illumina, San Diego, CA, USA) and imported into CLC Genomics Workbench12.0.4 (CLC bio, Aarhus, Denmark) for analysis. Trimmed reads were assembled into contigs by the de novo assembly command after they were quality trimmed.

### 2.3. Genome Analysis

The contigs were subjected to homology searches against the viral sequence dataset using the basic local alignment search tool (BLAST) (https://blast.ncbi.nlm.nih.gov/Blast.cgi accessed on 1 September 2022). The open reading frames (ORFs) were predicted by the Find open reading flame command in the CLC sequence viewer (CLC bio). The obtained genome sequences were aligned with the sequences deposited in the GenBank/EMBL/DDBJ database using ClustalW [17]. Pairwise sequence identity calculation was performed using CLC Genomics Workbench12.0.4 (CLC bio). Similarity plot and recombination analyses were performed using the SimPlot software v. 3.5.1 and Recombination Detection Program (RDP) v.4.80, respectively [18,19]. Phylogenetic analyses were performed on nucleotide sequences using the Neighbor-Joining method in MEGA7 [20]. The reliability of the phylogenetic tree cluster was evaluated by running 1000 replicates in the bootstrap analysis [21]. The GC content was calculated using VectorBuilder (https://www.vectorbuilder.jp/tool/gc-content-calculator.html accessed on 1 September 2022). For 3D protein structure modeling, AlphaFold v2.1.0 (Alphafold2) was employed in the Colab notebook (https://colab.research.google.com/github/deepmind/alphafold/blob/main/notebooks/AlphaFold.ipynb accessed on 1 September 2022) [22]. Structure representations from Alphafold2 were visualized using ChimeraX [23].

### 2.4. Identification of PAdVs from Fecal Samples of Wild Boar and Pigs Using PCR

To investigate the prevalence of PAdV, 113 rectal contents obtained from healthy wild boar hunted between 2018 and 2021 and 100 fecal samples collected from pigs (17 with diarrhea and 83 without diarrhea) between 2016 and 2020 were investigated. The total DNA was directly extracted from the supernatants of fecal samples diluted 1:9 (*w/v*) in MEM using the High Pure Viral Nucleic Acid Kit. The extracted nucleic acids were subjected to PCR using adenovirus consensus primers targeted for a conserved region of adenoviral DNA polymerase gene [24]. The PCR products were directly sequenced using the Sanger method. To further characterize PAdV-positive samples, two primer pairs targeted partial regions of the hexon and fiber genes (hexon: forward 5′-CCTACTTTGACATTCGGGGG-3′, reverse 5′- GGTGCTCTGAAGAACCATGTTG-3′, fiber: forward 5′- TCATTACCAACAGACTTTGATCC-3′, reverse 5′- CCCAACTAAAGGCTATTTGATAGTT-3′), which were designed from the nucleotide sequences of Ino5 and HNF-70. If expected size bands (hexon: 1364 bp, fiber: 1129 bp) were obtained, PCR products were directly sequenced using the Sanger method.

## 3. Results

### 3.1. Isolation and Identification of Porcine Adenovirus

A total of 11 fecal samples of wild boar were subjected to isolation of the virus. On a second passage at the third day post-inoculation, shrinking of some of the cells with clumping was observed in CPK cells inoculated in one wild boar sample (Figure 1). No CPE was observed in any cell lines apart from CPK cells. Thus, the supernatant of CPK cells subjected to deep sequencing and 32,403 bp length contig, which consisted of 578,377 specific reads in 691,638 total reads, was obtained. BlastN analysis revealed that the contig had sequence similarity of 97.95% to PAdV-5 Japanese porcine strain HNF-70 (AF289262) [13]. The near complete genome nucleotide sequence of PadV-5, named PadV-5/wild boar/JPN/Ino5/2020 (Ino5), was deposited in the DDBJ/EMBL/GenBank database under accession number LC702314.

### 3.2. Genome Analysis and Structural Modeling of the Hexon and Fiber Proteins

Open reading frames (ORF)s of Ino5 were predicted by the Find open reading flame command in the CLC sequence viewer (CLC bio, Aarhus, Denmark) and complete genome of Ino5 was compared with complete and partial sequences of other PAdV-5 strains. Complete genome sequencing of bovine adenovirus 2 KY19-1 (LC621239) that had second similarity with 71.51% (nt) to that of Ino5 was performed using similarity plot analysis in SimPlot software. Ino5 had the same genome organization with HNF-70 (Figure 2A) and exhibited approximately >93% nucleotide (nt) sequence homology to all other PAdV-5s throughout the whole genome, apart from the hexon and E3 612R, and fiber coding region (Figure 2B). Sequence nt and aa identities of the hexon, E3 612R, and fiber genes between Ino5 and HNF-70 were 89.58% and 91.34%, 74.64% and 65.18%, and 90.03% and 90.62%, respectively. Partial hexon and fiber to E4 coding region sequences of HuN02 and YN01 exhibited sequence homology to HNF-70 (Figure 2B). E3 612R and fiber gene region of HNF-61 showed sequence similarity (nt 98.0% and 96.28%, and 98.59% and 98.79%, respectively) to Ino5. Since these data suggest that there are recombination events between Ino5 and HNF-70, bootstrap scanning analysis using Ino5, HNF-70, and KY19-1 sequences by the RDP was performed. The result shows that two possible recombination breakpoints were identified in the hexon and E3 612R regions, supported by P-value cutoffs of 2.320 × 10^−26^, 1.890 × 10^−3^, and 5.724 × 10^−24^ for the RDP, GENECONV, and BootScan, respectively (the hexon region), and 5.545 × 10^−110^, 4.010 × 10^−71^, 2.081 × 10^−87^, 6.514 × 10^−36^, 9.955 × 10^−22^, 1.417 × 10^−43^, and 1.245 × 10^−1^ for the RDP, GENECONV, BootScan, MaxChi, Chimaera, SiScan, and 3Seq, respectively (the E3 612R region) (Figure 2C, Appendix A). These two possible recombination regions were found to be reduced in the GC nucleotide content under 50% (Figure 2D).

Since sequence differences of the hexon and fiber protein between Ino5 and HNF-70 were observed (Figure 2B), phylogenetic analyses using the full genome sequences of the hexon and fiber coding regions were performed. Ino5 and HNF-70 were distantly related, but formed a cluster in the hexon coding region (Figure 3A). In the fiber coding region tree, Ino5 branched with Japanese porcine PAdV-5 strain HNF-61, while HNF-70 branched with the Chinese porcine PadV-5 strain VIRES_HuN02 (MK377456) (Figure 3B). Next, the structures of the hexon and partial fiber proteins, including the head region of Ino5 and HNF-70, were predicted using Alphafold2 and compared to porcine adenovirus 3 (PAdV-3) (AC_000189) and human adenovirus 3 (HuAdV-3) GB strain (AY599834) (hexon and fiber), and PAdV-5VIRES_HuN02_C7 and PAdV-5 VIRES_YN01_C10 (MK377469) (fiber). The hexon monomer was divided into 11 fragments, which consist of two fragments in the tower region, five fragments in the neck region, and four fragments in the base region, based on variability and structure [25]. The prediction of the whole hexon protein structure revealed that PAdVs and HuAdV share structural similarities (Appendix A). The tower region of the adenovirus hexon, which consists of segments 3 and 7, is located on the top of the hexon (Figure 4A), contains neutralizing epitopes, and is the most variable region of the hexon [25]. Although partial sequence similarity between Ino5 and HNF-70 was observed in segments 3 and 7, hyper sequence difference was found mid-segment 3 (corresponding to the 170–190 hexon aa regions of Ino5) and the latter part of segment 7 (corresponding to the 396–412 hexon aa regions of Ino5) (Figure 4B), resulting in 3D-stractural difference between Ino5 and HNF-70 (Figure 4B). For the fiber protein, the head of fiber proteins corresponding to 367 to 496 aa positions of Ino5 shared high 3D-structural similarity to HNF-70 and HNF-61; however, structural differences between Ino5 and Chinese PAdV-5 strains VIRES_HuN02_C7 and VIRES_YN01_C10 were observed in latter part corresponding to 417 to 496 in Ino5 fiber protein positions (Appendix A).

### 3.3. Prevalence of PAdV in Wild Boars and Pigs, and Phylogenetic Analyses of the Hexon and Fiber Genes

PCR was employed to investigate the prevalence of adenoviruses using a primer pair targeting for the conserved region of the adenoviral DNA polymerase gene. Amplicons of approximately 300 base pairs were obtained from two wild boar samples (1.7%) and 12 pig samples (12.0%). All samples were from animals without diarrhea. These samples were subjected to amplification of the hexon and fiber genes using gene specific primer pairs. Ten partial hexon and nine partial fiber gene sequences were obtained. Phylogenetic analysis using partial nt sequence of the hexon gene revealed that all Japanese PAdVs were classified into the PAdV-5 group, and PAdV-5 strains formed two clusters (Figure 5A). One cluster consisted of three wild boar strains including Ino5 and eight porcine strains (colored in blue), and another cluster included HNF-70, Ebi-14/2021, and Ebi-17/2021 (colored in green). For the phylogenetic tree of partial nt sequences of the fiber gene, wild boar strains clustered with porcine strains that grouped together with wild boar strains in the hexon gene, except for four strains (Ebi-3, 5, 7, and 8/2021). Ebi-14/2021 grouped with HNF-70 in the hexon tree was blanched with wild boar strains. Ebi-3, 5, 7, and 8/2021 clustered with HFN-70 and Chinese PAdV-5 strain VIRES HuN02 (Figure 5B). Fiber gene sequencing of Ebi-17/2021 could not be obtained due to an insufficient amount of the sample.

## 4. Discussion

In this study, a PAdV-5 was isolated using CPK cells from wild boar for the first time. The natural host range of AdVs including cell culture has been thought to be restricted to a single host species [1]. We could not isolate PAdV using any cell lines other than pig origin. The PAdV-5 host range might be confined to pigs and wild boars. In the preliminary study on virus detection using wild boar fecal samples using a metagenomics approach, sufficient PAdV-5 specific sequence reads were not obtained. Metagenomic analysis using animal tissues or fecal samples usually result only in partial sequences [26,27]. Using the supernatant of virus cell cultures, a near complete genome sequence of Ino5 could be obtained by NGS. Genomic sequence data of PAdV-5 is scarce. Only one complete sequence (HNF-70), several partial sequences of porcine PadV-5 strains, including one Japanese strain (HNF-61) and three Chinese strains (HuN02, YN01_C2, and YN02_C1), and several short PAdV-5 sequences (<365 bp) from South Korea and India are available at present (as of September 2022). More sequence data and analyses are needed for a comprehensive recognition of PAdVs. Therefore, we performed sequence analyses of the PAdV-5 strain from wild boar and prevalence investigation of wild boars and pigs.

PAdV-5s were first isolated from pigs with respiratory symptoms in 1987 in Japan [6]. In this study, an intranuclear inclusion body was found in kidney medulla cells of a pig showing respiratory symptoms, which were positive for PAdV PCR via fecal sample and subjected to necropsy examination for the pathogenesis of the respiratory disease (Appendix A). However, an inclusion body was not identified in cells of the respiratory organs or intestinal tract. Although PAdV was reported as a cause of peritubular nephritis [28], the pig did not show nephritis. Thus, the relationship between PAdV-5 and respiratory disease, nephritis, or enteritis was unclear. PAdV is regarded as a low-grade pathogen in pigs because experiment PAdV infection of pigs shows no clinical signs except for mild diarrhea of short duration [2]. HuAdVs are important pathogens in humans, causing fatal acute respiratory disease in infants and adults who are immune compromised [29]. However, the majority of HuAdV infections in humans are also subclinical; therefore, recombinant human adenoviruses have been used for viral vectors of human vaccine delivery and gene therapy [30,31,32]. On the other hand, molecular evolution driven by recombination events influencing tropism and pathogenicity in a novel human adenovirus can cause epidemic diseases and continues evolution of new pathogens [33,34,35].

Whole genome of Ino5 was compared with whole genome of PAdV-5 HNF-70 and partial genome obtained from GenBank database. HuAdV-D is the largest HuAdV species. HuAdV-Ds share >90% of highly conserved genomes; however, there are specific loci of genetic hypervariability in the penton base, hexon, fiber, and E3 CR1α, β, and γ genes [29]. Like HuAdV-Ds, Ino5 has approximately >93% nt sequence homology to all other PAdV-5s throughout the whole genome, apart from the hexon and E3 612R to fiber coding region. Complete genome nt and aa sequences of the penton base of Ino5 exhibit high sequence identities (99.36% nt and 99.36% aa) to those of HNF-70, even though it was isolated more than 35 years ago. The nt sequence identities of the hexon, E3 612R, and fiber genes between Ino5 and HNF-70 were lower than other regions (89.58%, 74.64%, and 90.03%, respectively), as is the case with HuAdVs. E3 612R of Ino5 exhibits the most sequence differences from that of HNF-70. The E3 gene product provides the means for subverting the host defense mechanisms [36]; however, at least 60% of the E3 region of PAdV-5 is not essential for virus replication [37]. Fowl AdVs are classified based on the highly variable region of the L1 loop in the hexon gene and, to a lesser extent, in the fiber gene of the virus [38]. Although the PAdV-5 genotype was not examined due to insufficient sequence information, sequence variability of the hexon and fiber was found in this study. Homologous recombination among capsid genes (the penton base, hexon, and fiber) is essential for obtaining the high sequence diversity of AdVs [29,39,40,41,42]. In this study, two possible recombination breakpoints were identified in the hexon and E3 612R regions using RDP. GC content is related to genome stability and protection of recombination [43,44]. In fact, two possible recombination regions in this study were located where GC content was reduced under 50%.

Hexon protein is the main neutralizing antigen of AdVs and serotype-specific antigens are located mainly on the tower region [25,45]. Structural comparison of the hexon of Ino5 and HNF-70 using the Alphafolds2 reveals that, although structural homology was found among PAdVs and HuAdV-3, structural differences between Ino5 and HNF-70 were identified in the tower region corresponding to segments 3 and 7, suggesting antigenic differences between these strains. In the structural comparison of the fiber head, Ino5, HNF-70, and HNF-61 share structural similarity with each other; however, structural differences were found in the latter part of the fiber head between Japanese PAdV-5 strains and Chinese PAdV-5 strains, VIRES_HU_N02_C7 and VIRES_YN01_C10. HNF-70 and VIRES_HU_N02_C7 are closely related to each other in the fiber phylogenetic tree (Figure 3B) and share 95.05% nt sequence identity. However, aa sequence identity of the fiber region between HNF-70 and VIRES_HU_N02_C7 was 91.93%. These findings suggest that Japanese and Chinese PAdVs may have evolved independently with antigenic mutation by non-synonymous substitution. Fiber also induces host immunity like the hexon, and is an important determinant of neutralization and hemagglutination-inhibition via antibodies [29]. Structural differences of the fibers suggest antigenic differences between Japanese and Chinese strains.

Prevalence investigation using PCR targeting for the adenoviral DNA polymerase gene shows that 1.7% wild boar samples and 12.0% pig samples were positive. Sequence analysis reveals that all detected AdVs were classified into PAdV-5. Epidemiological survey and genotyping PAdV are limited. In the survey using pooled fecal samples from pigs in Spain, 70% (17/24) samples were positive for PAdV, and PAdV-3 was the predominant genotype (98%) [43]. In the survey in Thailand, the prevalence of PAdV infection in pigs was 16.9% (132/781) and most of PAdVs were PAdV-3 [12]. On the other hand, it has been reported that PAdV-5 was a predominant genotype detected in river water in New Zealand and Spain [46,47,48]. Hirahara et al. performed a serological survey on PAdV in pigs in Japan and reported that, although neutralizing antibodies against all serotypes (PAdV-1 to PAdV-5) were detected, the positive rate of PAdV-5 (>80%) was higher than other serotypes [11]. Thus, PAdV-5 may have been a predominant serotype of pigs since at least 1980 in Japan. Phylogenetic analyses using partial hexon and fiber genes shows that some strains were grouped into different clusters in these trees, suggesting homologous recombination events. To our knowledge, although a recombination event between PAdV and atadenovirus in the fiber gene has been reported [49], this study presents the first report on homologous recombination among PAdV-5s.

## 5. Conclusions

A PAdV-5 was isolated using CPK cells from fecal samples of wild boar for the first time. Sequence analysis reveals that this strain, Ino5, has sequence homology to the Japanese PAdV-5 strain HNF-70 that was isolated 35 years ago; however, sequence differences were detected in the hexon, E3 612R, and fiber genes, suggesting homologous recombination events. Structural analysis shows that partial structural differences between Ino5 and other PAdV-5 strains suggests antigenic heterogeneity. These data imply that PAdV-5 has evolved with homologous recombination events in the same manner as human adenoviruses among not only pig populations, but also in wild boars in Japan. This information provides important information on the genetic plasticity and evolution of animal AdVs.

## Figures and Tables

**Figure 1 viruses-14-02400-f001:**
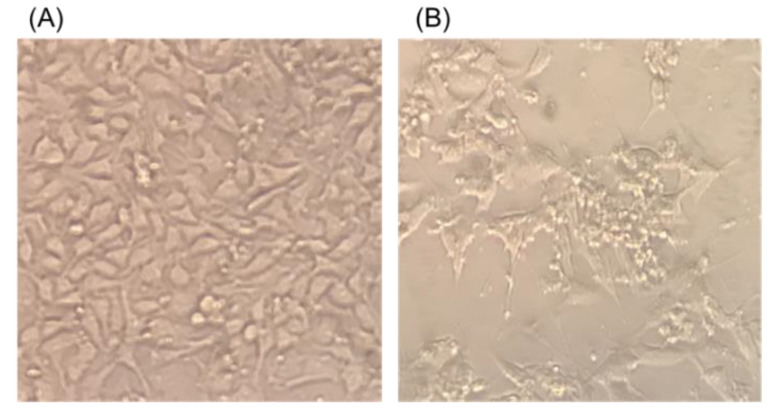
T Morphology of uninfected CPK cells (**A**) and 72 h after infection (**B**) with PAdV-5/WB/Ino5/2020/JPN at eighth passages (×200).

**Figure 2 viruses-14-02400-f002:**
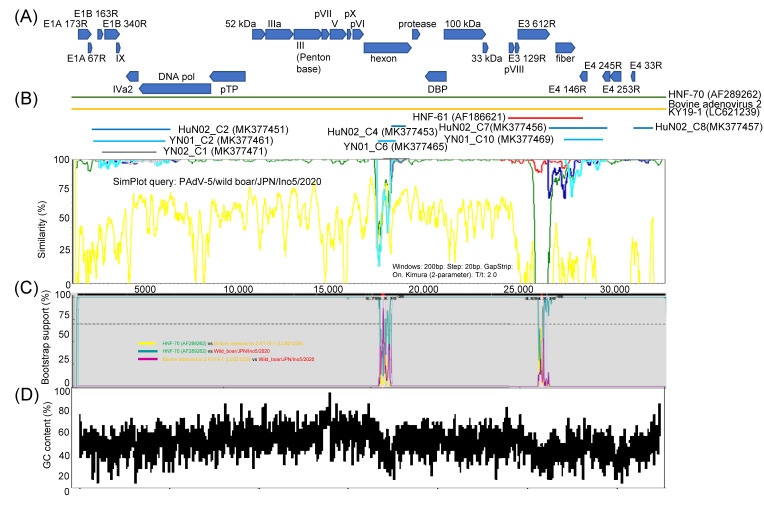
T Schematic illustration of the genome organization of PAdV-5/WB/Ino5/2020/JPN. Blue arrows indicate open reading frames (**A**) Similarity plot of the whole genome of PAdV-5 HNF-70 (green curve), bovine adenovirus 2 KY-19-1 (orange curve), partial genome of PAdV-5 HNF-61 (red curve), PAdV-5 HuN02 (blue curves), PAdV-5 YN01 (light blue curves), YN02 (grey curves), and PAdV-5 Ino5 as query sequence, with a sliding window of 200 nt and a moving step size of 20 nt (**B**) Recombination analysis of PAdV-5 HNF-70 vs. bovine adenovirus 2 KY19-1 (yellow curve), PAdV-5 HNF-70 vs. PAdV-5 Ino5 (green curve), and bovine adenovirus 2 KY19-1 vs. PAdV-5 Ino5 (purple curve) (**C**) Average GC% content per gene across PAdV-5 Ino5 genome (**D**).

**Figure 3 viruses-14-02400-f003:**
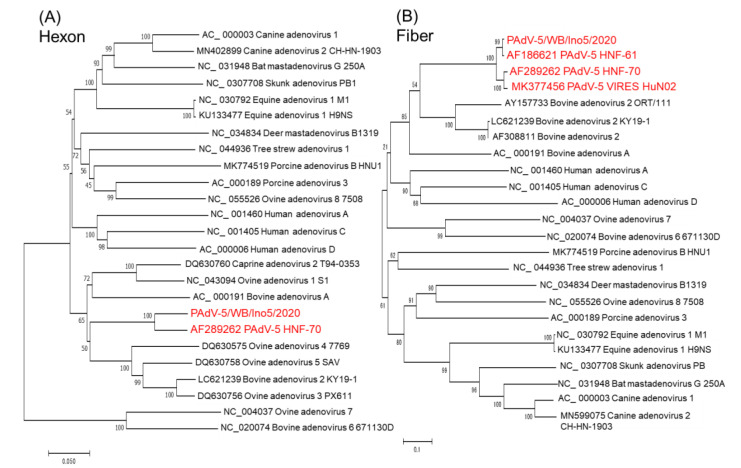
Phylogenetic analysis based on the full genome nucleotide sequences of the hexon (**A**) and fiber (**B**) coding region of PAdV-5s and other adenoviruses. The trees were constructed according to the neighbor-joining method in MEGA7 with 1000 bootstrap values. The bar represents a corrected genetic distance. PAdV-5 strains are colored in red.

**Figure 4 viruses-14-02400-f004:**
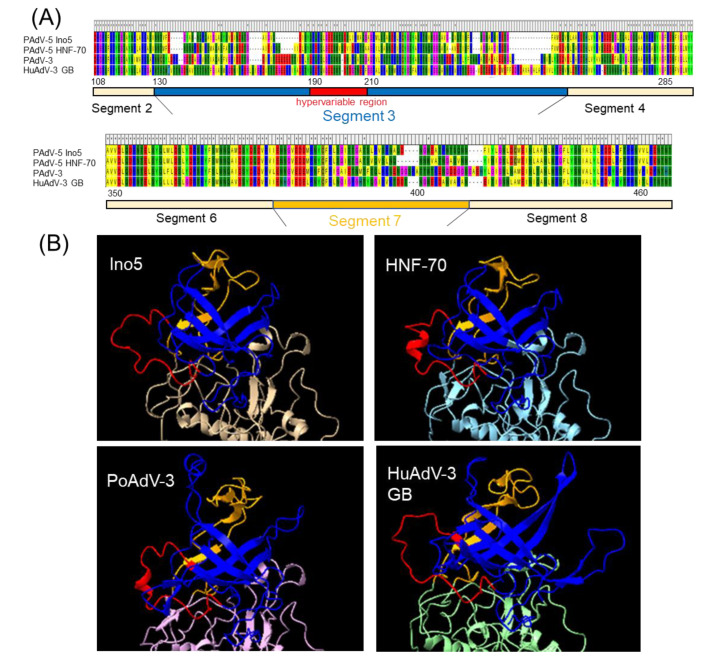
Alignment of aa partial hexon protein sequences of PAdV-5 Ino5, PAdV-5 HNF-70 (AF289262), porcine adenovirus 3 (AC_000189), and Human adenovirus 3 GB strain (AY599834) (**A**) Structure prediction and comparison of the hexon tower region of PAdV-5 Ino5, PAdV-5 HNF-70, porcine adenovirus 3, and Human adenovirus 3 GB strain. The three-dimensional protein structures were predicted by Alphafold2 (https://colab.research.google.com/github/sokrypton/ColabFold/blob/main/AlphaFold2.ipynb accessed on 1 September 2022) and visualized in ChimeraX. The predicted 3D structure of the tower region of the hexon protein segment 3, hypervariable region in the segment 3, and segment 7 are colored by blue, red, and orange (**B**).

**Figure 5 viruses-14-02400-f005:**
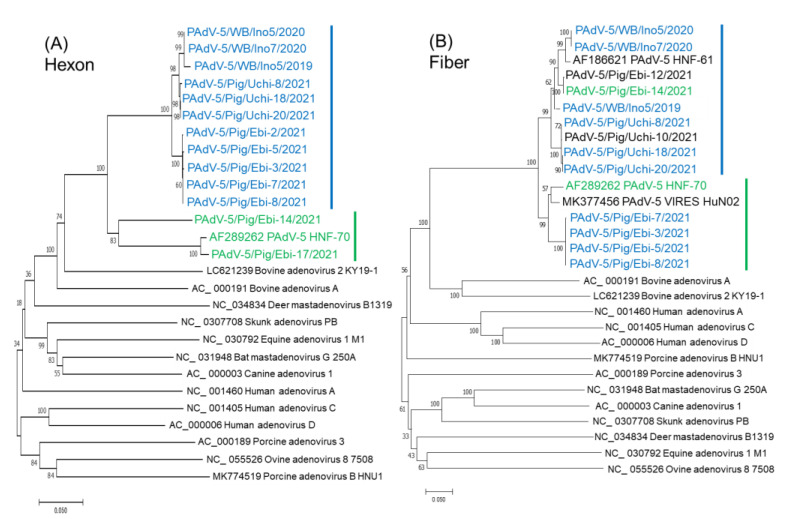
Phylogenetic analysis based on partial nucleotide sequences of the hexon (**A**) and fiber (**B**) coding region of PAdV-5s and other adenoviruses. The trees were constructed according to the neighbor-joining method in MEGA7 with 1000 bootstrap values. The bar represents a corrected genetic distance. Strains clustered with Ino5 and HNF-70 in the hexon tree were colored in blue and green, respectively.

## Data Availability

The original contributions presented in the study are included in the article/Appendix A. The raw reads generated in this study are available through the corresponding author upon request.

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
