# Peer review of "First Isolation and Identification of Homologous Recombination Events of Porcine Adenovirus from Wild Boar"

_viruses, 2022, doi:10.3390/v14112400_

Round 1
Reviewer 1 Report
Overall, this is a straightforward description of the first identification of porcine adenovirus (PAdV) in a wild boar. Identification began with virus isolation from a fecal sample. Sequence analysis showed identity with PAdV5. However, three regions showed less identity, the result of possible recombination events. Variation occurred in immunologically relevant regions in the proteome. This paper has important implications in variability within PAdV5 genomes and the potential for immunological escape. The results are also interesting considering the potential role of wild boar as a source for new PAdV5 peptide sequences.
Author Response
Thank you very much for your kind comments and suggestions on our manuscript. We appreciated your comments and suggestions, and revised our manuscript in line with Reviewer 2’s reports. I thank you again for your kind attention.

Reviewer 2 Report
Overall Comments:
Porcine adenoviruses (PAdVs) are classified within the genus Mastadenovirus in the Adenoviridae family, and are regarded as low grade pathogens, infecting the porcine populations worldwide. In Japan, PAdV-5 strains were first isolated from pigs with in 1987. In this manuscript, Oba and co-authors isolated a PAdV-5 strain [PAdV-5/wild boar/JPN/Ino5/2020 (Ino5)] from a fecal sample of wild boar by using CPK cells for the first time, and reported that the PAdV-5 strain could not isolate PAdV using any cell lines other than pig origin. Subsequently, deep sequencing of the Ino5 strain was performed based on the cDNA libraries construction, and the whole-genome sequence of PAdV-5/wild boar/JPN/Ino5/2020 (Ino5) was deposited in the DDBJ/EMBL/GenBank database under accession number LC702314. In addition, the whole-genome analysis show that the isolated PAdV-5 strain (Ino5) has approximately >93% sequence homology throughout the genome with the PAdV-5 Japanese porcine strain HNF-70 (AF289262), which was isolated in Japan in 1987, and other PAdV-5s strains except the hexon and E3 612R, and fiber coding region. Complete genome analysis revealed that the PAdV-5 has sequence homology to other PAdV-5 strains and two recombination breakpoints in the hexon and E3 612R regions were detected. Finally, they concluded that PAdV-5 has evolved with homologous recombination events in the same manner as human adenoviruses.
Overall, the manuscript is well written, the methods described in good detail, and the figures with corresponding legends provide the data in a clear form. The conclusions of the study are supported by the data presented, and are clearly stated. However, several points need to be addressed before the manuscript can be recommended for publication.
Minor comments:
1) Line 61, what’s the cell lines’ ATCC No.
2) Line 118, Remove the extra spaces before the word “Ino5”.
3) Line 180, Even though the phylogenetic analysis based on partial nucleotide sequences of the hexon and fiber coding region of PAdV-5s and other adenoviruses was performed in this study, it is still necessary to use the full genome sequence for a systemic phylogenetic analysis to analyze the evolution relationship between the virus strains isolated from wild boars and those isolated from other countries (for example, HNF-70, HNF-61 VIRES_HuN02_C7 and VIRES_YN01_C10). The new results can be organized as figure 4. This can make the analysis more comprehensive and the conclusion more convincing.
4) Line 225-228, What animal hosts were the Asian PAdV-5 strains isolated from?
5) The structural differences between Ino5 and Chinese PAdV-5 strains VIRES_HuN02_C7 and VIRES_YN01_C10 were observed in this study. What is the phylogenetic and evolutionary relationship between them?
Author Response
Thank you very much for your kind comments and suggestions on our manuscript. We appreciated your comments and suggestions, and revised our manuscript in line with your reports. Our reactions to your comments and suggestions are as follows.
Comment 1): Line 61, what’s the cell lines’ ATCC No.
Response: We added the cell line’s Japanese Collection of Research Bioresources (JCRB) No. or cited related references (line 61-62).
Comment 2): Line 118, Remove the extra spaces before the word “Ino5”.
Response: We removed the extra space before the word “Ino5” in lines 118.
Comment 3): Line 180, Even though the phylogenetic analysis based on partial nucleotide sequences of the hexon and fiber coding region of PAdV-5s and other adenoviruses was performed in this study, it is still necessary to use the full genome sequence for a systemic phylogenetic analysis to analyze the evolution relationship between the virus strains isolated from wild boars and those isolated from other countries (for example, HNF-70, HNF-61 VIRES_HuN02_C7 and VIRES_YN01_C10). The new results can be organized as figure 4. This can make the analysis more comprehensive and the conclusion more convincing.
Response: In accordance with your suggestion, we constructed phylogenetic trees using the full genome sequences of the hexon and fiber coding regions as figure 3 (Fig. 3 of the old version has been renamed Fig. 4) and added two sentences in the Results section (line 170-174). Unfortunately, only one complete hexon coding region sequence (HNF-70) and three complete fiber coding region sequences (HNF-70, HNF-61, and VIRES_HuN02_C7 (however, about 90 nt 5’ terminal sequences of this strain are missing)) are available. Approximately 620 nt sequences of 5’ terminal of VIRES_YN01_C10 fiber region (the shaft region of the fiber) are missing; thus, VIRES_YN01_C10 was omitted in the phylogenetic analysis. These results are discussed in the Discussion section according to your comment (Comment 5).
Comment 4): Line 225-228, What animal hosts were the Asian PAdV-5 strains isolated from?
Response: All Asian PAdV-5 strains were isolated from swine. This information has been included in line 245.
Comment 5): The structural differences between Ino5 and Chinese PAdV-5 strains VIRES_HuN02_C7 and VIRES_YN01_C10 were observed in this study. What is the phylogenetic and evolutionary relationship between them?
Response: According to your suggestion, we added two sentences concerning the phylogenetic and evolutionary relationship between Japanese and Chinese PAdVs “HNF-70 and VIRES_HU_N02_C7 were closely related to each other in the fiber phylogenetic tree (Fig. 3B) and shared 95.05% nt sequence identity. However, aa sequence identity of the fiber region between HNF-70 and VIRES_HU_N02_C7 was 91.93%. These finding suggest that Japanese and Chinese PAdVs may have evolved independently with anti-genic mutation by non-synonymous substitution.” (line 297-302).

Round 2
Reviewer 2 Report
The authors reply all my comments. The manuscript is revised nicely. I recommend the paper should be accepted for publication.
Author Response
Thank you very much for your kind comments and suggestions on our manuscript. We appreciated your comments and suggestions, and revised our manuscript in line with academic editor’s comment. I thank you again for your kind attention.